# Characteristics of aspiration pneumonia patients in acute care hospitals: A multicenter, retrospective survey in Northern Japan

Jun Suzuki[1], Ryoukichi Ikeda[1]*, Kengo Kato[1], Risako Kakuta[1], Yuta Kobayashi[1], Akira Ohkoshi[1], Ryo Ishii[1], Ai Hirano-Kawamoto[1], Jun Ohta[1], Rei Kawata[2], Tomonori Kanbayashi[1], Masaki Hatano[1], Tadahisa Shishido[1], Yuya Miyakura[3], Kento Ishigaki[4], Yasunari Yamauchi[5], Miho Nakazumi[6], Takuya Endo[7], Hiroki Tozuka[8], Shiori Kitaya[2], Yuki Numano[6], Shotaro Koizumi[4], Yutaro Saito[2], Mutsuki Unuma[4], Ken Hashimoto[7], Eiichi Ishida[8], Toshiaki Kikuchi[8], Takayuki Kudo[3], Kenichi Watanabe[4], Masaki Ogura[5], Masaru Tateda[8], Takatsuna Sasaki[6], Nobuo Ohta[2], Tatsuma Okazaki[9], Yukio Katori[1]

1 Department of Otolaryngology and Head and Neck Surgery, Tohoku University Graduate School of Medicine, Sendai, Japan, 2 Division of Otolaryngology, Tohoku Medical and Pharmaceutical University Hospital, Sendai, Japan, 3 Department of Otolaryngology, South Miyagi Medical Center, Ogawara, Japan, 4 Department of Otolaryngology, Tohoku Rosai Hospital, Sendai, Japan, 5 Department of Otolaryngology, Sendai City Hospital, Sendai, Japan, 6 Department of Otolaryngology, Osaki Citizen Hospital, Osaki, Japan, 7 Department of Otolaryngology, Japanese Red Cross Ishinomaki Hospital, Ishinomaki, Japan, 8 Department of Otolaryngology, Sendai Medical center, Sendai, Japan, 9 Department of Physical Medicine and Rehabilitation, Tohoku University graduate School of Medicine, Sendai, Japan

* ryoukich@hotmail.com

**Data Availability Statement:** All relevant data are within the paper and its Supporting Information files.

## Abstract

### Background

Pneumonia is a common cause of illness and death of the elderly in Japan. Its prevalence is escalating globally with the aging of population. To describe the latest trends in pneumonia hospitalizations, especially aspiration pneumonia (AP) cases, we assessed the clinical records of pneumonia patients admitted to core acute care hospitals in Miyagi prefecture, Japan.

### Methods

A retrospective multi-institutional joint research was conducted for hospitalized pneumonia patients aged ≥20 years from January 2019 to December 2019. Clinical data of patients were collected from the medical records of eight acute care hospitals.

### Results

Out of the 1,800 patients included in this study, 79% of the hospitalized pneumonia patients were aged above 70 years. The most common age group was in the 80s. The ratio of AP to total pneumonia cases increased with age, and 692 out of 1,800 patients had AP. In univariate analysis, these patients had significantly older ages, lower body mass index (BMI), a lower ratio of normal diet intake and homestay before hospitalization, along with more AP

**Funding:** This research was supported by AMED under Grant Number 19dk0310101h0001.

**Competing interests:** The authors have declared that no competing interests exist.

recurrences and comorbidities. During hospitalization, AP patients had extended fasting periods, more swallowing assessments and interventions, longer hospitalization, and higher in-hospital mortality rate than non-AP patients. A total of 7% and 2% AP patients underwent video endoscopy and video fluorography respectively. In multivariate analysis, lower BMI, lower C-reactive protein, a lower ratio of homestay before hospitalization, a higher complication rate of cerebrovascular disease, dementia, and neuromuscular disease were noted as a characteristic of AP patients. Swallowing interventions were performed for 51% of the AP patients who had been hospitalized for more than two weeks. In univariate analysis, swallowing intervention improved in-hospital mortality. Lower AP recurrence before hospitalization and a lower ratio of homestay before hospitalization were indicated as characteristics of AP patients of the swallowing intervention group from multivariate analysis. Change in dietary pattern from normal to modified diet was observed more frequently in the swallowing intervention group.

## Conclusion

AP accounts for 38.4% of all pneumonia cases in acute care hospitals in Northern Japan. The use of swallowing evaluations and interventions, which may reduce the risk of dysphagia and may associate with lowering mortality in AP patients, is still not widespread.

## 1. Introduction

Pneumonia is a globally common infection and a major cause of death in adults [1, 2]. Its risks increase with age, and it is the main cause of hospitalization and mortality among the elderly [2, 3]. Review papers show that aspiration is strongly implicated in pneumonia in the elderly, and aspiration pneumonia (AP) is estimated to account for 5%-15% cases of community-acquired pneumonia (CAP), but the information of the rates of AP in hospital-acquired pneumonia (HAP) is limited [1, 4]. AP is a multi-factorial condition; impaired swallowing, abnormal cough reflex, host immune defense, and pathogen factors are intricately involved [1, 4]. The reported risk factors of AP are: age, male sex, dysphagia, diabetes mellitus, degenerative neurologic and lung diseases, impaired consciousness and dementia, dehydration, and use of antipsychotic drugs and proton pump inhibitors [5–9]. According to a report from Japan in 2008, the incidence of AP in CAP and HAP cases was 60.1% and 86.7%, respectively, and AP accounted for 66.8% of total hospitalized pneumonia patients [10]. The aging rate of the Japanese population (aged ≥65 years) was high at 28.1% in 2018, making Japan the world's leading country in confronting healthcare problems of the elderly [11]. Pneumonia, including AP, was the third leading cause of death (9.7%) in Japan in 2018 [12]. The study that reported the high incidence of AP in CAP and HAP [10] was conducted a decade ago. The understanding of the relationship between AP and its risk factors, such as dysphagia, has recently improved. In addition, people in Japan have started to recognize AP as a common and fatal disease. Therefore, we believe that the ratio of AP has changed in the current social environment. Moreover, the epidemiology of pneumonia and the clinical features and outcomes of AP remain largely unknown.

In order to understand the current diagnosis and treatment of pneumonia in the elderly, we investigated (1) the incidence of hospitalized pneumonia according to age, (2) the rate of AP in hospitalized pneumonia patients, (3) the rate and characteristics of swallowing

intervention-received pneumonia patients, (4) the occupations of the people who evaluated swallowing function and their methods, (5) evaluation of the effect of swallowing interventions on in-hospital mortality, (6) causative microorganisms in recent AP cases, and (7) the recent use of antibiotics, by assessing the clinical records of patients admitted to core acute care hospitals in Miyagi Prefecture, Japan, during the year 2019.

## 2. Materials and methods

### 2.1 Participants and data collection

A total of 1,800 pneumonia patients above 20 years of age, who were admitted to all the departments of (1) South Miyagi Medical Center (92 cases), (2) Tohoku University Hospital (107 cases), (3) Tohoku Rosai Hospital (278 cases), (4) Sendai City Hospital (149 cases), (5) Sendai Medical Center (78 cases), (6) Tohoku Medical and Pharmaceutical University Hospital (388 cases), (7) Osaki Citizen Hospital (289 cases), (8) Japanese Red Cross Ishinomaki Hospital (419 cases) in Miyagi Prefecture (Fig 1), Japan, between January 2019 and December 2019, were enrolled in the study. Miyagi prefecture has a total population of 2.3 million, including both urban (Sendai area: 1.5 million people) and rural zones (Sennan area: 180,000 people, Osaki/Kurihara area: 270,000 people, Ishinomaki/Tome/Kesennuma area: 350,000 people, Fig 1). Acute care hospitals participating in this study were core general hospitals of each area with emergency departments. These hospitals provided medical services to patients with CAP, nursing- and healthcare-associated pneumonia (NHCAP), corresponding to health-care-associated pneumonia (HCAP) [13, 14], and HAP. Exclusion criteria were patients with interstitial pneumonia, fungal and/or acid-fast bacilli infection, acquired immunodeficiency syndrome (AIDS), and transplantation.

Clinical data including age, sex, body mass index (BMI), vital signs, laboratory data, comorbidities, residential and feeding status, microbiological examinations, use of antibacterial agents, alternative nutrition during hospitalization, swallowing assessments and interventions, and clinical outcomes were collected from medical records of each hospital. Clinical outcomes

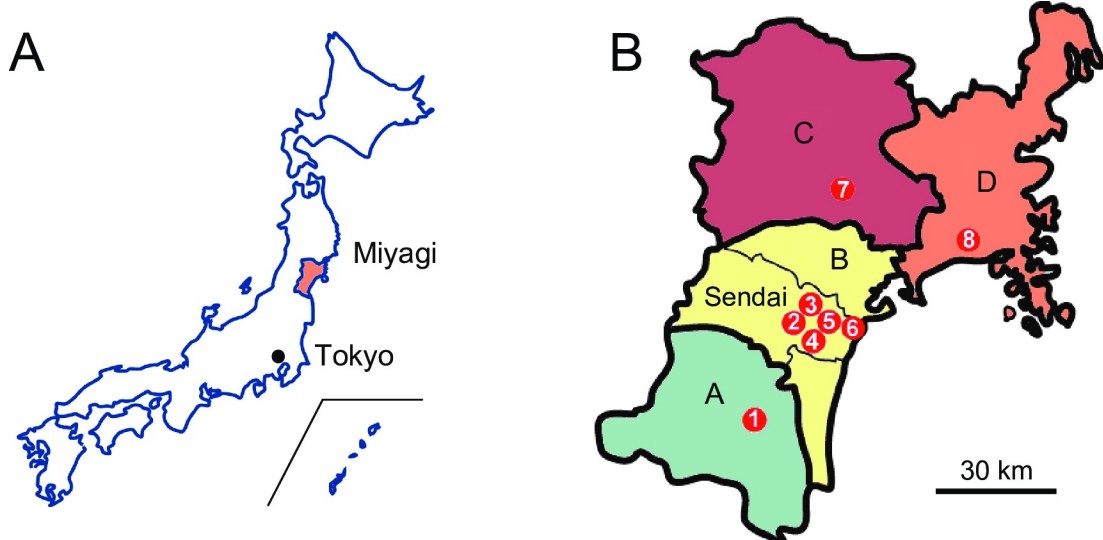

**Fig 1. Localization of participating hospitals in Miyagi prefecture, Japan.** A: a map of Japan. B: a map of Miyagi Prefecture. (1) South Miyagi Medical Center, (2) Tohoku University Hospital, (3) Tohoku Rosai Hospital, (4) Sendai City Hospital, (5) Sendai Medical Center, (6) Tohoku Medical and Pharmaceutical University Hospital, (7) Osaki Citizen Hospital, (8) Japanese Red Cross Ishinomaki Hospital. A: Sennan area, B: Sendai area, C: Osaki/Kurihara area, D: Ishinomaki/Tome/Kesennuma area.

were evaluated by length of hospitalization and in-hospital mortality. This study was approved by the Tohoku University Hospital Institutional Review Board (IRB) (Reference number: 2019-1-971). All data were fully anonymized, and IRB in our institution waived the requirement for informed consent.

## 2.2 Criteria of pneumonia

Pneumonia was defined as an acute illness associated with the presence of new pulmonary infiltrates on chest radiograph and/or computed tomography (CT), along with at least two or three respiratory symptoms (cough, sputum production, wheezing, fever, and dyspnea) and inflammation determined by blood analyses of white blood cell (WBC) count and C-reactive protein (CRP). The diagnostic criteria for pneumonia were determined by a guideline from the Japan Respiratory Society. The definition of AP was based on that of the Japanese Study Group on Aspiration Pulmonary Disease: (1) pneumonia with an overt aspiration witnessed by surrounding persons; (2) pneumonia with a strongly suspected aspiration condition; (3) pneumonia with a predisposition to aspiration because of abnormal swallowing function or dysphagia, (4) compatible radiologic findings of aspiration pneumonia [15]. Pneumonia patients who did not meet the criteria were defined as patients with non-aspiration pneumonia (non-AP).

## 2.3 Definition of swallowing intervention

The Japanese Society of Dysphagia Rehabilitation described "Summaries of training methods in 2014" which contained 25 indirect and 13 direct exercises [16]. We categorized pneumonia patients who received one and more kinds of swallowing exercises into swallowing intervention group.

## 2.4 Microbiological and antibiotic studies

We defined CAP as pneumonia in patients at home before admission and NHCAP/HAP as pneumonia in patients in nursing homes or other hospitals and analyzed the microbiological and antibiotic characteristics of AP and non-AP patients in each CAP and NHCAP/HAP group. Standard microbiological procedures were performed for investigation of pathogens in two sets of blood and/or sputum samples. Antibody testing for *Mycoplasma pneumoniae* and *Chlamydia pneumoniae* and urinary antigen detection for *Streptococcus pneumoniae* and *Legionella pneumophila* was done, if indicated, by the attending physician. The number of antibiotics, if more than one was consumed, was counted.

## 2.5 Statistical analysis

The Student's t-test, the Wilcoxon–Mann–Whitney test, or Fisher's exact test was performed using statistical software SPSS version 27 (IBM, Chicago, IL, USA) or JMP Pro version 15 (SAS Institute Inc., Cary, NC, USA). Factors characterizing AP and swallowing intervention group were determined using logistic regression analysis with independent variables with presumed clinical importance by using JMP Pro version 15. Considering the covariance, we selected independent variables regarding factors characterizing aspiration pneumonia as follows: general status (age, sex); nutritional condition (BMI, diet before hospitalization); inflammation severity (C-reactive protein, SpO2); past medical history (cerebrovascular disease, dementia, neuromuscular disease); residential status; assessment and treatment (fasting, swallowing intervention); outcomes (length of hospitalization, mortality). Selected independent variables regarding factors characterizing swallowing intervention group were as follows: general status

(age, sex); nutritional condition (BMI); inflammation severity (C-reactive protein, SpO2); past medical history (cerebrovascular disease, dementia, neuromuscular disease, AP recurrence); residential status; assessment and treatment (fasting, swallowing assessment); outcomes (length of hospitalization, mortality). Differences with a corrected p-value of less than 0.05 were considered significant. Data were presented as mean ± standard deviation.

## 3. Result

### 3.1 Aspiration pneumonia vs. non-aspiration pneumonia

Overall, 1,800 patients met the inclusion criteria and were included in this study (Fig 2). A total of 79% of the hospitalized pneumonia patients were aged above 70 years, and the most common age group was in the 80s (Fig 3A). The ratio of AP to total pneumonia cases increased with age (Fig 3B). Out of 1,419 pneumonia patients aged above 70 years, 607 patients (42.8%) were diagnosed with AP compared to 85 of 381 patients under 70 years of age (22.3%). Patient

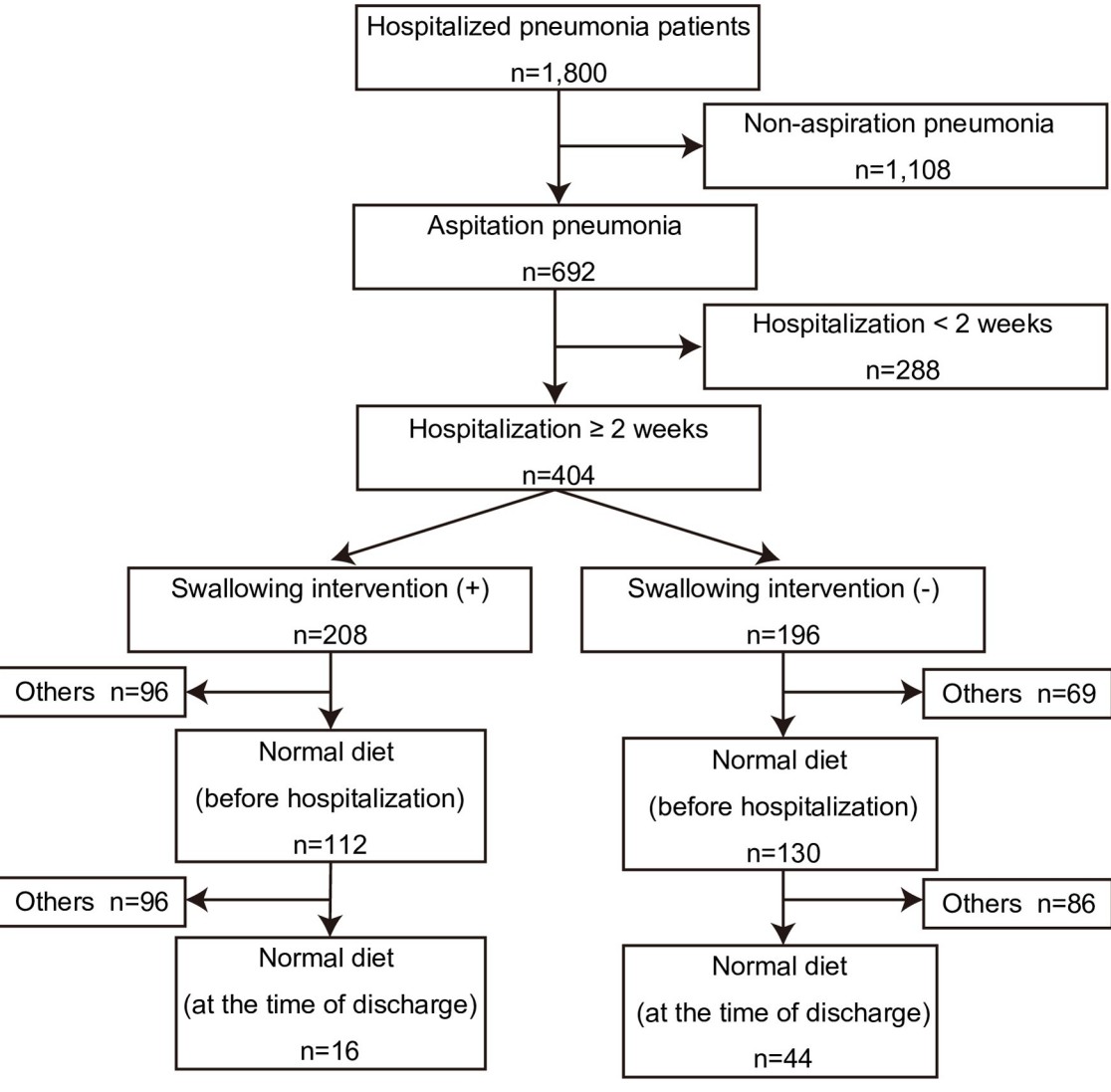

**Fig 2. A flow chart of patients evaluated throughout the course of this study.**

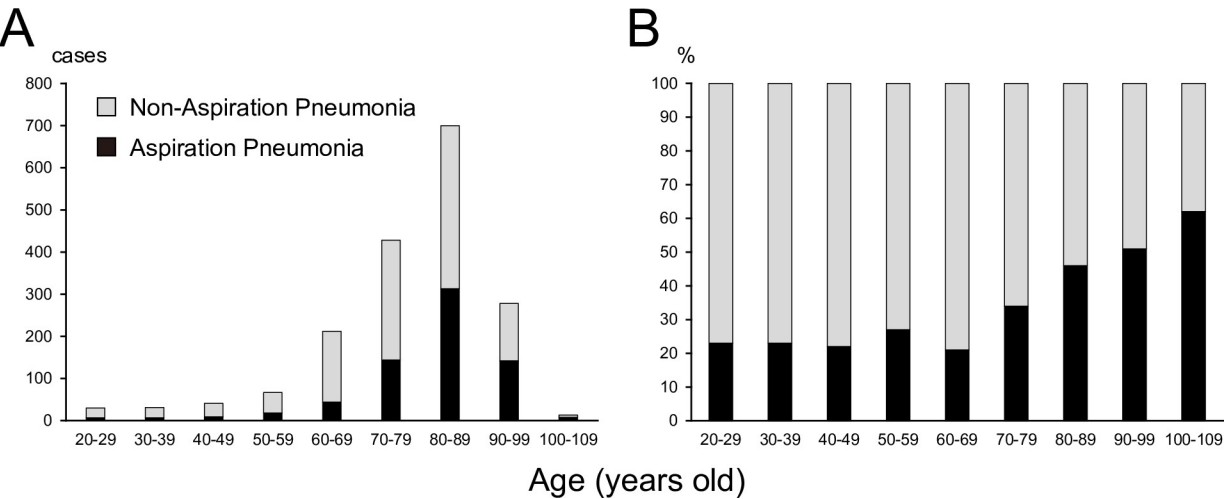

**Fig 3. The ratio of aspiration pneumonia to total pneumonia cases.** A: The number of hospitalized aspiration and non-aspiration pneumonia patients according to age. B: The percentages of aspiration and non-aspiration pneumonia patients in total hospitalized pneumonia patients according to age.

characteristics of both AP and non-AP groups are shown in Table 1. AP group had 692 cases (38.4%) and non-AP group had 1,108 cases (61.6%). The mean age was significantly higher in AP group compared with non-AP group. On the contrary, BMI, CRP, serum albumin, and oxygen saturation under room air measured at admission were significantly lower in AP group than in non-AP group. The rates of AP recurrences and comorbidities such as dementia, cerebrovascular, and neuromuscular diseases were significantly higher in AP group than in non-AP group.

We divided diet into two types, normal diet or diet modified to help to swallow. The rates of normal diet intake and homestay before hospitalization were significantly lower in the AP group than in the non-AP group (Table 1). The AP group had a longer fasting period and a higher rate of alternative nutrient use, such as applying a nasogastric tube and parenteral nutrition than the non-AP group. Speech therapists and nurses mainly assessed dysphagia, and speech therapists and nutritional support teams mainly performed swallowing intervention in both AP and non-AP patients. Detailed swallowing evaluations through video endoscopy and videofluoroscopic evaluation were performed in 7% (51 cases) and 2% (14 cases) AP patients, respectively. Length of hospitalization and mortality rate was significantly higher in AP group than in the non-AP group (Table 1). At the time of discharge, 68% of non-AP patients took a normal diet, whereas only 23% of patients with AP took a normal diet (Table 1). Correspondingly, the rates of use of modified diet, gastrostomy, and nasogastric tube were significantly higher in patients with AP (total 63%) than in the non-AP patients (total 30%).

To reveal the characteristics of AP patients compared with non-AP patients, we performed a multivariate analysis. Logistic regression analysis showed that lower BMI, lower C-reactive protein, a lower ratio of homestay before hospitalization, a higher complication rate of cerebrovascular disease, dementia, and neuromuscular disease, more fasting controls, and more swallowing interventions were noted as a characteristic of AP patients (Table 2).

### 3.2 Characteristics of swallowing intervention cases

Some patients with AP received swallowing interventions, while others did not (Table 1). In order to clarify the characteristics of AP patients with swallowing interventions, we also

**Table 1. Characteristics of patients with aspiration and non-aspiration pneumonia.**

| | Aspiration pneumonia | Non-aspiration pneumonia | P Value |
|---|---|---|---|
| **Patients** | 692 (38%) | 1108 (62%) | |
| Age | 81.2 ±12.6 | 75.4 ±15.0 | < 0.001 |
| Sex (female) | 251 (36%) | 400 (36%) | 0.861 |
| Body mass index, mean ± SD | 19.4 ±4.0 | 21.4 ±4.6 | < 0.001 |
| Body temperature | 37.6 ±1.5 | 37.7 ±1.2 | 0.145 |
| **Laboratory data** | | | |
| Leukocyte counts | 11119.5 ±6348.5 | 10552.6 ±5790.7 | 0.070 |
| C-reactive protein (mg/dl) | 9.4 ±8.5 | 11.5 ±9.8 | < 0.001 |
| Serum albumin (g/dl) | 3.1 ±0.6 | 3.2 ±0.6 | < 0.001 |
| SpO2 breathing room air on admission | 91.7 ±7.0 | 93.1 ±6.0 | < 0.001 |
| **Comorbidity** | | | |
| Cancer | 187 (27%) | 314 (28%) | 0.543 |
| Cerebrovascular disease | 206 (30%) | 153 (14%) | < 0.001 |
| Dementia | 270 (39%) | 124 (11%) | < 0.001 |
| Neuromuscular disease | 93 (13%) | 34 (3%) | < 0.001 |
| Chronic respiratory disease | 192 (28%) | 418 (38%) | < 0.001 |
| AP recurrence case | 240 (35%) | 304 (27%) | 0.002 |
| **Residential status before hospitalization** | | | |
| Home | 471 (68%) | 1003 (91%) | < 0.001 |
| The other hospital | 53 (8%) | 23 (2%) | < 0.001 |
| Nursing home | 166 (23%) | 73 (7%) | < 0.001 |
| **Oral intake before hospitalization** | | | |
| Normal diet | 397 (57%) | 953 (86%) | < 0.001 |
| Modified diet | 235 (34%) | 119 (11%) | < 0.001 |
| Gastrostomy | 38 (6%) | 16 (1%) | < 0.001 |
| Nasogastric tube | 14 (2%) | 10 (1%) | 0.042 |
| **Treatment, Evaluation, Intervention** | | | |
| Fasting | 573 (83%) | 410 (37%) | < 0.001 |
| Fasting period (day) | 8.3 ±12.0 | 2.6 ±8.2 | < 0.001 |
| Alternative nutrition during hospitalization | | | |
| Nasogastric tube | 74 (11%) | 39 (4%) | < 0.001 |
| Total parenteral nutrition | 39 (6%) | 40 (4%) | 0.052 |
| Peripheral parenteral nutrition | 620 (90%) | 692 (62%) | < 0.001 |
| Oral intake only | 34 (5%) | 316 (29%) | < 0.001 |
| Swallowing assessment | 410 (59%) | 294 (27%) | < 0.001 |
| Video Endoscopy | 51 (7%) | 19 (2%) | < 0.001 |
| Video fluorography | 14 (2%) | 3 (0%) | 0.002 |
| Speech therapist | 245 (35%) | 135 (12%) | < 0.001 |
| Nurse | 275 (40%) | 184 (17%) | < 0.001 |
| Swallowing intervention | 264 (38%) | 140 (13%) | < 0.001 |
| Speech therapist | 225 (33%) | 118 (11%) | < 0.001 |
| Nurse | 21 (3%) | 15 (1%) | 0.023 |
| Dentist | 45 (7%) | 28 (3%) | < 0.001 |
| Nutritional support team | 76 (11%) | 29 (3%) | < 0.001 |
| **Outcomes** | | | |
| Length of hospitalization (day) | 22.7 ±24.1 | 17.0 ±17.7 | < 0.001 |
| Mortality | 119 (17%) | 95 (9%) | < 0.001 |

*(Continued)*

**Table 1.** (Continued)

| | Aspiration pneumonia | Non-aspiration pneumonia | P Value |
|---|---|---|---|
| **Oral intake at discharge** | | | |
| Normal diet | 129 (23%) | 684 (68%) | < 0.001 |
| Modified diet | 273 (48%) | 266 (26%) | < 0.001 |
| Gastrostomy | 47 (8%) | 21 (2%) | < 0.001 |
| Nasogastric tube | 41 (7%) | 17 (2%) | < 0.001 |

SD; standard deviation, SpO2; saturation of percutaneous oxygen, AP; aspiration pneumonia

evaluated AP patients with hospitalization of 14 days and longer periods because it was difficult to evaluate the effect of swallowing interventions on short-time hospitalization cases. Patient characteristics of the swallowing intervention and non-swallowing intervention group were shown in Table 3. A total of 404 patients met the inclusion criteria, and 208 patients received swallowing interventions (51%). In the swallowing intervention group, the mean age and rate of patients with cerebrovascular disease were significantly higher, fasting period was longer, and oxygen saturation under room air, rate of patients with neuromuscular disease, and AP recurrence rate were significantly lower compared with non-swallowing intervention groups. Correspondingly, the rate of normal diet intake before hospitalization was significantly lower in the swallowing intervention group. The number of patients with the above three comorbidities was 157 in each group. Speech therapists mainly performed swallowing assessments and interventions in 86% of cases, including overlapping.

Detailed swallowing evaluations through video endoscopy and video fluorography were performed in 20% (41 cases) and 5% (11 cases) of patients with swallowing intervention, respectively. Regarding outcomes of patients, although fasting periods and lengths of hospitalization were significantly longer, mortality was significantly lower in the swallowing intervention group (8% vs. 16%, p = 0.0178).

To reveal the characteristics of swallowing intervention cases, we performed a multivariate analysis. Logistic regression analysis showed that a lower ratio of AP recurrence, a lower ratio

**Table 2. Factors characterizing aspiration pneumonia by logistic regression analysis.**

| | Coefficient | SE | P Value | Odds ratio | 95% CI |
|---|---|---|---|---|---|
| Constant | -1.650 | 1.342 | | | |
| Age | -0.010 | 0.006 | 0.079 | 1.010 | 0.999–1.022 |
| Sex (female) | 0.104 | 0.080 | 0.196 | 0.812 | 0.592–1.113 |
| Body mass index | 0.077 | 0.019 | < .0001 | 0.926 | 0.891–0.961 |
| C-reactive protein | 0.021 | 0.009 | 0.015 | 0.979 | 0.962–0.996 |
| SpO2 breathing room air on admission | 0.001 | 0.012 | 0.946 | 0.999 | 0.975–1.023 |
| Cerebrovascular disease | -0.241 | 0.096 | 0.012 | 1.619 | 1.113–2.355 |
| Dementia | -0.434 | 0.098 | < .0001 | 2.384 | 1.621–3.507 |
| Neuromuscular disease | -0.493 | 0.162 | 0.002 | 2.681 | 1.422–5.055 |
| Residential status before hospitalization (home) | -0.349 | 0.105 | 0.001 | 0.498 | 0.330–0.751 |
| Fasting | -0.744 | 0.083 | < .0001 | 4.432 | 3.203–6.034 |
| Swallowing intervention | -0.508 | 0.093 | < .0001 | 2.760 | 1.915–3.978 |
| Length of hospitalization | 0.000 | 0.003 | 0.962 | 1.000 | 0.993–1.001 |
| Mortality | -0.015 | 0.132 | 0.911 | 1.030 | 0.614–1.727 |

SE; standard error, 95%CI; 95% ofconfidence interval, SpO2; saturation of percutaneous oxygen, AP; aspiration pneumonia

**Table 3. Characteristics of aspiration pneumonia patients with swallowing interventions.**

| | Swallowing intervention (+) | Swallowing intervention (-) | P Value |
|---|---|---|---|
| **Patients** | 208 (51%) | 196 (49%) | |
| Age | 83.3 ±9.9 | 79.5 ±12.4 | < 0.001 |
| Sex (female) | 73 (35%) | 67 (34%) | 0.917 |
| Body mass index, mean ± SD | 19.2 ±4.1 | 18.8 ±4.2 | 0.292 |
| Body temperature | 37.6 ±2.0 | 37.7 ±1.2 | 0.629 |
| **Laboratory data** | | | |
| Leukocyte counts | 10610.9 ±4704.9 | 10980.7 ±5830.9 | 0.923 |
| C-reactive protein (mg/dl) | 10.4 ±9.0 | 10.4 ±8.6 | 0.840 |
| Serum albumin (g/dl) | 3.0 ±0.6 | 3.1 ±0.6 | 0.595 |
| SpO2 breathing room air on admission | 90.6 ±8.3 | 92.9 ±5.5 | 0.008 |
| **Comorbidity** | | | |
| Cancer | 54 (26%) | 57 (29%) | 0.483 |
| Cerebrovascular disease | 79 (38%) | 48 (25%) | 0.004 |
| Dementia | 88 (42%) | 72 (37%) | 0.253 |
| Neuromuscular disease | 22 (11%) | 35 (18%) | 0.038 |
| Chronic respiratory disease | 50 (24%) | 58 (30%) | 0.219 |
| AP recurrence case | 56 (27%) | 74 (38%) | 0.020 |
| **Residential status before hospitalization** | | | |
| Home | 138 (66%) | 136 (69%) | 0.514 |
| The other hospital | 19 (9%) | 17 (9%) | 0.871 |
| Nursing home | 51 (25%) | 41 (21%) | 0.389 |
| **Oral intake before hospitalization** | | | |
| Normal diet | 112 (54%) | 130 (66%) | 0.011 |
| Modified diet | 89 (43%) | 46 (23%) | < 0.001 |
| Gastrostomy | 3 (1%) | 20 (10%) | < 0.001 |
| Nasogastric tube | 7 (3%) | 3 (1%) | 0.230 |
| **Treatment, Evaluation, Intervention** | | | |
| Fasting | 189 (91%) | 164 (84%) | 0.030 |
| Fasting period (day) | 13.3 ±16.4 | 10.1 ±11.8 | 0.024 |
| Alternative nutrition during hospitalization | | | |
| Nasogastric tube | 35 (17%) | 25 (13%) | 0.251 |
| Total parenteral nutrition | 25 (12%) | 12 (6%) | 0.040 |
| Peripheral parenteral nutrition | 199 (96%) | 174 (89%) | 0.009 |
| Oral intake only | 4 (2%) | 12 (6%) | 0.031 |
| Swallowing assessment | 200 (96%) | 78 (40%) | < 0.001 |
| Video Endoscopy | 41 (20%) | 6 (3%) | < 0.001 |
| Video fluorography | 11 (5%) | 3 (1%) | 0.004 |
| Speech therapist | 178 (86%) | 15 (8%) | < 0.001 |
| Nurse | 111 (53%) | 58 (30%) | < 0.001 |
| Swallowing intervention | 208 (100%) | 0 (0%) | < 0.001 |
| Speech therapist | 179 (86%) | 0 (0%) | < 0.001 |
| Nurse | 16 (8%) | 0 (0%) | < 0.001 |
| Dentist | 38 (18%) | 0 (0%) | < 0.001 |
| Nutritional support team | 66 (32%) | 0 (0%) | < 0.001 |
| **Outcomes** | | | |
| Length of hospitalization (day) | 34.7 ±25.9 | 32.3 ±27.5 | 0.026 |
| Mortality | 17 (8%) | 31 (16%) | 0.018 |

(*Continued*)

**Table 3.** (*Continued*)

| | Swallowing intervention (+) | Swallowing intervention (-) | P Value |
|---|---|---|---|
| **Oral intake at discharge** | | | |
| Normal diet | 18 (9%) | 48 (29%) | < 0.001 |
| Modified diet | 120 (63%) | 67 (41%) | < 0.001 |
| Gastrostomy | 10 (5%) | 23 (14%) | 0.005 |
| Nasogastric tube | 20 (10%) | 15 (9%) | 0.663 |

SD; standard deviation, SpO$_2$; saturation of percutaneous oxygen, AP; aspiration pneumonia

of homestay before hospitalization, a higher rate of swallowing assessments were noted as characteristics of swallowing intervention cases in AP patients (Table 4).

From the swallowing intervention and non-swallowing intervention group, a total of 112 (54%) and 130 (66%) patients respectively had a normal diet intake before hospitalization, and 16 (14%) and 44 (34%) patients respectively had a normal diet intake at the time of discharge (Table 5).

## 3.3 Causative organisms and choice of antibiotics

There were significant differences between the causative organisms of the AP and non-AP groups in CAP, with a higher frequency of *Staphylococcus aureus*, *Klebsiella* spp., and *Escherichia coli*, and a lower frequency of *S. pneumoniae*, and *Haemophilus influenzae* in the AP group (Table 6). The detection rate of anaerobic bacteria, e.g., *Bacteroides* spp., *Prevotella* spp., and *Fusobacterium* spp. was low in both groups.

In HNCAP/HAP, there were no significant differences between AP and non-AP groups as to the causative organisms (Table 7).

**Table 4. Factors characterizing swallowing intervention group by logistic regression analysis.**

| | Coefficient | SE | P Value | Odds ratio | 95% CI |
|---|---|---|---|---|---|
| Constant | -0.780 | 3.255 | | | |
| Age | -0.029 | 0.018 | 0.097 | 1.030 | 0.995–1.067 |
| Sex (female) | 0.018 | 0.182 | 0.920 | 0.964 | 0.472–1.970 |
| Body mass index | 0.067 | 0.047 | 0.154 | 0.935 | 0.852–1.026 |
| C-reactive protein | 0.001 | 0.021 | 0.946 | 0.999 | 0.959–1.040 |
| SpO2 breathing room air on admission | 0.031 | 0.028 | 0.269 | 0.970 | 0.999–1.031 |
| Cerebrovascular disease | 0.266 | 0.196 | 0.175 | 1.702 | 0.789–3.670 |
| Dementia | -0.101 | 0.193 | 0.602 | 0.817 | 0.383–1.744 |
| Neuromuscular disease | 0.050 | 0.305 | 0.870 | 1.105 | 0.334–3.658 |
| AP recurrence case | -0.562 | 0.184 | 0.002 | 0.325 | 0.158–0.670 |
| Residential status before hospitalization (home) | -0.429 | 0.208 | 0.039 | 0.424 | 0.188–0.959 |
| Normal diet before hospitalization | -0.266 | 0.192 | 0.166 | 0.587 | 0.277–1.248 |
| Fasting | -0.471 | 0.292 | 0.106 | 0.390 | 0.124–1.223 |
| Swallowing assessment | 2.002 | 0.308 | < .0001 | 54.840 | 16.389–183.506 |
| Length of hospitalization | -0.015 | 0.008 | 0.063 | 1.015 | 0.999–1.031 |
| Mortality | -0.001 | 0.300 | 0.999 | 0.999 | 0.308–3.243 |

SE; standard error, 95%CI; 95% ofconfidence interval, SpO$_2$; saturation of percutaneous oxygen, AP; aspiration pneumonia

**Table 5. The rate of normal diet intake in initially normal diet-taking aspiration pneumonia patients after swallowing interventions.**

|  | Normal diet | Modified diet | No oral intake / Unknown | Death |
|---|---|---|---|---|
| Swallowing intervention (+) | 16 (14.3%) | 64 (57.1%) | 23 (20.5%) | 9 (8.0%) |
| Swallowing intervention (-) | 44 (33.8%) | 43 (33.1%) | 20 (15.4%) | 23 (17.7%) |

In CAP, the most frequent empirical antibiotic treatment in AP group was Sulbactam / Ampicillin (SBT/ABPC) (64.5% in AP group vs. 36.5% in non-AP group, $p<0.001$). Clindamycin (3.2% vs 0.8%, $p = 0.001$) was also more frequently used in AP group. Conversely, Ceftriaxone (CTRX) (20.0% vs. 37.3%, $p<0.001$) was more frequently used in non-AP group. Levofloxacin (LVFX) (4.0% vs. 11.1%, $p<0.001$) was also more frequently used in non-AP group (Table 8).

**Table 6. Pathogen distribution in community-acquired pneumonia (CAP).**

| CAP | Aspiration pneumonia (n = 471) | | Non-aspiration pneumonia (n = 1003) | | P value |
|---|---|---|---|---|---|
|  | N |  | N |  |  |
| *Streptococcus pneumoniae* | 38 | 8.2% | 152 | 15.3% | < 0.001 |
| penicillin-susceptible | 34 |  | 121 |  |  |
| penicillin-resistant | 2 |  | 6 |  |  |
| *Staphylococcus aureus* | 91 | 19.7% | 100 | 10.1% | < 0.001 |
| methicillin-susceptible | 59 |  | 67 |  |  |
| methicillin-resistant | 32 |  | 33 |  |  |
| *Pseudomonas aeruginosa* | 27 | 5.8% | 37 | 3.7% | 0.075 |
| *Klebsiella spp.* | 76 | 16.4% | 60 | 6.0% | < 0.001 |
| ESBL | 5 |  | 4 |  |  |
| Non-ESBL | 63 |  | 45 |  |  |
| *Escherichia coli* | 30 | 6.5% | 30 | 3.0% | 0.003 |
| ESBL | 11 |  | 6 |  |  |
| Non-ESBL | 19 |  | 19 |  |  |
| *Haemophilus influenzae* | 21 | 4.5% | 81 | 8.2% | 0.011 |
| BLNAS | 4 |  | 19 |  |  |
| BLPAR | 2 |  | 8 |  |  |
| BLNAR | 10 |  | 21 |  |  |
| BLPACR | 0 |  | 4 |  |  |
| *Moraxella catarrhalis* | 12 | 2.6% | 30 | 3.0% | 0.738 |
| *Proteus spp.* | 3 | 0.6% | 4 | 0.4% | 0.686 |
| ESBL | 0 |  | 1 |  |  |
| Non-ESBL | 3 |  | 2 |  |  |
| *Mycoplasma pneumoniae* | 1 | 0.2% | 1 | 0.1% | 0.535 |
| *Legionella spp.* | 2 | 0.4% | 12 | 1.2% | 0.248 |
| *Bacteroides spp.* | 2 | 0.4% | 1 | 0.1% | 0.239 |
| *Prevotella spp.* | 0 | 0.0% | 2 | 0.2% | 1.000 |
| *Fusobacterium spp.* | 0 | 0.0% | 1 | 0.1% | 1.000 |
| Others | 58 | 12.5% | 149 | 15.0% | 0.227 |
| Unknown | 191 | 41.3% | 444 | 44.8% | 0.195 |

ESBL; Extended spectrum β-lactamase, BLNAS; β-lactamase non-producing ampicillin susceptible, BLPAR; β-lactamase producing ampicillin resistant, BLNAR; β-lactamase-non-producing ampicillin-resistance, BLPACR; β-lactamase producing amoxicillin/clavulanic acid resistant.

**Table 7. Pathogen distribution in nursing- and healthcare-associated pneumonia and hospital-acquired pneumonia (NHCAP/HAP).**

| NHCAP/HAP | Aspiration pneumonia | | Non-aspiration pneumonia | | P value |
|---|---|---|---|---|---|
| | (n = 219) | | (n = 96) | | |
| | N | | N | | |
| *Streptococcus pneumoniae* | 19 | 8.8% | 12 | 12.6% | 0.309 |
| *penicillin-susceptible* | 18 | | 7 | | |
| *penicillin-resistant* | 0 | | 2 | | |
| *Staphylococcus aureus* | 59 | 27.3% | 18 | 18.9% | 0.120 |
| *methicillin-susceptible* | 22 | | 5 | | |
| *methicillin-resistant* | 37 | | 13 | | |
| *Pseudomonas aeruginosa* | 22 | 10.2% | 5 | 5.3% | 0.192 |
| *Klebsiella spp.* | 26 | 12.0% | 11 | 11.6% | 1.000 |
| *ESBL* | 1 | | 1 | | |
| *Non-ESBL* | 17 | | 7 | | |
| *Escherichia coli* | 20 | 9.3% | 7 | 7.4% | 0.667 |
| *ESBL* | 11 | | 6 | | |
| *Non-ESBL* | 8 | | 0 | | |
| *Haemophilus influenzae* | 6 | 2.8% | 3 | 3.2% | 1.000 |
| *BLNAS* | 1 | | 1 | | |
| *BLPAR* | 1 | | 1 | | |
| *BLNAR* | 2 | | 0 | | |
| *BLPACR* | 1 | | 0 | | |
| *Moraxella catarrhalis* | 6 | 2.8% | 1 | 1.1% | 0.680 |
| *Proteus spp.* | 4 | 1.9% | 0 | 0.0% | 0.317 |
| *ESBL* | 2 | | 0 | | |
| *Non-ESBL* | 2 | | 0 | | |
| *Mycoplasma pneumoniae* | 0 | 0.0% | 0 | 0.0% | 1.000 |
| *Legionella spp.* | 1 | 0.5% | 0 | 0.0% | 1.000 |
| *Bacteroides spp.* | 1 | 0.5% | 0 | 0.0% | 1.000 |
| *Prevotella spp.* | 0 | 0.0% | 0 | 0.0% | 1.000 |
| *Fusobacterium spp.* | 0 | 0.0% | 0 | 0.0% | 1.000 |
| Others | 20 | 9.3% | 13 | 13.7% | 0.239 |
| Unknown | 78 | 36.1% | 38 | 40.0% | 0.612 |

ESBL; Extended spectrum β-lactamase, BLNAS; β-lactamase non-producing ampicillin susceptible, BLPAR; β-lactamase producing ampicillin resistant, BLNAR; β- lactamase-non-producing ampicillin-resistance, BLPACR; β-lactamase producing amoxicillin/clavulanic acid resistant.

In NHCAP/HAP, there were no significant differences between AP and non-AP groups as to the selection of antibiotics: SBT/ABPC, CTRX, and Tazobactam/Piperacillin were frequently used in both groups (Table 9).

## 4. Discussion

In this study, we evaluated 1,800 hospitalized pneumonia patients in acute care hospitals. A total of 79% of the hospitalized patients with pneumonia were above 70 years of age, and the most common age group was in the 80s. From the univariate analyses, we revealed as follows: (1) AP patients showed significantly older ages, lower BMI, more numbers of comorbidities, more AP recurrences, lesser normal diet intake, and lesser homestay before hospitalization

**Table 8. Selection of antibiotics in community-acquired pneumonia (CAP).**

| CAP | Aspiration pneumonia (n = 470) | | Non-aspiration pneumonia (n = 986) | | P Value |
|---|---|---|---|---|---|
| | N | % | N | % | |
| Ampicillin | 3 | 0.6% | 7 | 0.7% | 1.000 |
| Penicillin G | 0 | 0.0% | 1 | 0.1% | 1.000 |
| Sulbactam / Ampicillin | 303 | 64.5% | 360 | 36.5% | < 0.001 |
| Piperacillin | 5 | 1.1% | 12 | 1.2% | 1.000 |
| Tazobactam / Piperacillin | 56 | 11.9% | 142 | 14.4% | 0.220 |
| Cefotaxime | 0 | 0.0% | 6 | 0.6% | 0.186 |
| Ceftriaxone | 94 | 20.0% | 368 | 37.3% | < 0.001 |
| Cefotiam | 1 | 0.2% | 1 | 0.1% | 0.542 |
| Imipenem / Cilastatin | 0 | 0.0% | 1 | 0.1% | 1.000 |
| Meropenem | 20 | 4.3% | 63 | 6.4% | 0.116 |
| Panipenem / Betamipron | 0 | 0.0% | 0 | 0.0% | 1.000 |
| Biapenem | 1 | 0.2% | 2 | 0.2% | 1.000 |
| Levofloxacin | 19 | 4.0% | 109 | 11.1% | < 0.001 |
| Ciprofloxacin | 0 | 0.0% | 4 | 0.4% | 0.312 |
| Pazufloxacin | 1 | 0.2% | 0 | 0.0% | 0.323 |
| Minocycline | 4 | 0.9% | 28 | 2.8% | 0.013 |
| Clindamycin | 15 | 3.2% | 8 | 0.8% | 0.001 |
| Vancomycin | 5 | 1.1% | 8 | 0.8% | 0.767 |
| Linezolid | 2 | 0.4% | 0 | 0.0% | 0.104 |
| Amikacin | 0 | 0.0% | 1 | 0.1% | 1.000 |
| Sulfamethoxazole—Trimethoprim | 0 | 0.0% | 5 | 0.5% | 0.182 |

compared with non-AP patients; (2) patients with AP had more extended fasting periods, more swallowing assessment and intervention rates, longer hospitalization, and higher in-hospital mortality rate during hospitalization. Multivariate analyses showed that lower BMI, lower C-reactive protein, a lower ratio of homestay before hospitalization, a higher complication rate of cerebrovascular disease, dementia, and neuromuscular disease were significant characteristics of AP patients. These results might reflect the current situation of hospitalized pneumonia in acute care hospitals in Japan, the country possessing the most aged society in the world.

## 4.1 AP in aged society

We showed that AP accounts for 38.4% (692 in 1,800) of all hospitalized pneumonia cases, including those of CAP, NHCAP, and HAP, and 42.8% (607 in 1,419) cases in the elderly (aged ≥70 years). As seen in a previous report [10], this study also shows that the rate of AP among total pneumonia cases increases with age in patients above 70 years (Fig 1B). AP comprises 22.3% of pneumonia patients under 70 years of age. This result might be due to the inclusion of young patients with severe complications such as malignancies, neuromuscular, and respiratory diseases. The presence of these patients may reflect the current situation of acute care hospitals.

In this study, the most common age group for AP was in the '80s (Fig 1A), and this peak of age group had shifted 10 years older compared with the previous report from Japan [10]. Since aging is accompanied by the onset of pneumonia [17], this result might reflect the dramatic increase in the number of aged people (≥65 years) from 28.22 million (22.1% of the total population in Japan) in 2008 to 35.56 million (28.1%) in 2018.

**Table 9. Selection of antibiotics in nursing- and healthcare-associated pneumonia and hospital-acquired pneumonia (NHCAP/HAP).**

| NHCAP/HAP | Aspiration pneumonia (n = 212) | | Non-aspiration pneumonia (n = 96) | | P Value |
|---|---|---|---|---|---|
| | N | % | N | % | |
| Ampicillin | 2 | 0.9% | 1 | 1.0% | 1.000 |
| Penicillin G | 0 | 0.0% | 1 | 1.0% | 0.323 |
| Sulbactam / Ampicillin | 123 | 58.0% | 48 | 47.5% | 0.216 |
| Piperacillin | 2 | 0.9% | 2 | 2.0% | 0.597 |
| Tazobactam / Piperacillin | 38 | 17.9% | 12 | 11.9% | 0.249 |
| Cefotaxime | 2 | 0.9% | 0 | 0.0% | 1.000 |
| Ceftriaxone | 40 | 18.9% | 26 | 25.7% | 0.133 |
| Cefotiam | 1 | 0.5% | 0 | 0.0% | 1.000 |
| Imipenem / Cilastatin | 0 | 0.0% | 0 | 0.0% | 1.000 |
| Meropenem | 19 | 9.0% | 3 | 3.0% | 0.060 |
| Panipenem / Betamipron | 0 | 0.0% | 0 | 0.0% | 1.000 |
| Biapenem | 0 | 0.0% | 0 | 0.0% | 1.000 |
| Levofloxacin | 3 | 1.4% | 4 | 4.0% | 0.210 |
| Ciprofloxacin | 0 | 0.0% | 0 | 0.0% | 1.000 |
| Pazufloxacin | 0 | 0.0% | 0 | 0.0% | 1.000 |
| Minocycline | 1 | 0.5% | 5 | 5.0% | 0.015 |
| Clindamycin | 6 | 2.8% | 2 | 2.0% | 1.000 |
| Vancomycin | 2 | 0.9% | 4 | 4.0% | 0.088 |
| Linezolid | 1 | 0.5% | 0 | 0.0% | 1.000 |
| Amikacin | 0 | 0.0% | 0 | 0.0% | 1.000 |
| Sulfamethoxazole–Trimethoprim | 1 | 0.5% | 1 | 1.0% | 0.542 |

Compared to the result of a study in 2008 that shows that AP accounts for 66.8% (394 in 589) of all pneumonia cases, and 80.1% (306 in 382) of elderly pneumonia cases (aged ≥70 years) [10], the percentage of AP cases in our study (38.4% and 42.8%, respectively) was apparently low. Recent studies from Japan reported that the prevalences of AP were 18.2% (adult) [18], 46.7% (aged ≥15 years) [19], and 50.4% (aged ≥70 years) [20]. Although robust diagnostic criteria for AP are lacking and surveys on AP consequently include heterogeneous patient populations [1], these results imply that the prevalence of AP has fallen over the last decade. This phenomenon might be partly caused by the increased awareness of dysphagia that can result in AP. Thus, the current medical personnel might be more aware of AP than those a decade ago. In fact, in acute care hospitals in Miyagi prefecture, the number of swallowing evaluations in 2019 was greater than those in the middle of the 2000 decade because of the change in the otolaryngology system in this region. These changes may reduce the ratio of AP in total pneumonia in older people.

## 4.2 Clinical features and outcomes of AP

Aspiration is widely recognized as an important risk factor for pneumonia [21], and dysphagia affects up to 60% of the institutionalized elderly population [22–24]. However, there are limited data on the clinical characteristics of AP available at present [8, 9, 19, 25]. Using systematic literature review, van der Maarel-Wierink et al. identified 13 risk factors of AP: age, male sex, lung diseases, dysphagia, diabetes mellitus, dementia, angiotensin I-converting enzyme deletion/deletion genotype, bad oral health, malnutrition, Parkinson's disease, and the use of antipsychotic drugs, proton pump inhibitors, and angiotensin-converting enzyme inhibitors [9]. Accordingly, in this study, the patients with AP were older and showed a greater prevalence of

dementia, neuromuscular diseases, and malnutrition than non-AP patients. We diagnosed malnutrition by low BMI and low serum albumin. Other observational studies indicated many potential risk factors of AP [8, 19, 25, 26]; decrease in BMI, and presence of confusion (dementia), cerebrovascular and neuromuscular disease was also confirmed from the multivariate analysis in our study. Compared with non-AP patients, AP patients in this study had a more extended hospitalization and a higher in-hospital mortality rate. These results were consistent with previous reports [19, 25, 27]. A recent systematic review of AP patients with CAP also suggests that AP is associated with both higher in-hospital and 30-day mortality rates in CAP patients outside the intensive care unit [28]. A number of aspiration risk factors, e.g., dementia, poor performance status, and sleeping drugs, are associated with a rise in mortality and recurrence of pneumonia in the elderly [8]. Future multicenter prospective studies will be needed to identify new risk factors for AP.

### 4.3 Swallowing assessment and intervention of hospitalized pneumonia patients

The recurrence of pneumonia is characteristic of AP [19, 25, 27]. Considering this fact, many AP patients had modified diets or gastric gavage feedings before hospitalization. In this study, swallowing evaluations were performed for 59.2% (410/692 cases) of AP patients, and 39.1% (704/1,800 cases) of total pneumonia patients. Speech-language pathologists and nurses mainly performed swallowing evaluations. Otolaryngologists mainly performed video endoscopy and videofluoroscopic evaluation in 7% of AP patients and 2% of non-AP patients. Recent reports showed that risk-adjusted mortality of AP patients is lower in hospitals reporting a high frequency of AP than in hospitals reporting a low frequency of AP [27]. It must be important to pick up appropriate candidates for swallowing assessments to find undiagnosed dysphagia patients.

### 4.4 Effects of swallowing intervention

Among AP patients who were hospitalized for more than two weeks, 51% of the patients received swallowing interventions (208/404 patients). Patients in the swallowing intervention group had older ages, longer fasting periods and hospitalization, but had a significantly lower mortality rate than those in the non-intervention group in the univariate analyses. However, the multivariate analysis did not support these results. Since this study was a retrospective cross-sectional study, future studies are required to investigate the effects of swallowing intervention on the outcomes of AP patients.

The present study revealed that swallowing interventions significantly changed the dietary pattern from a normal diet to a modified diet. Malnutrition is common in elderly people, especially those with chronic disorders [29], and is associated with poor hospitalization outcomes [30]. Modified diets have been considered to decrease mis-swallowing [31, 32] and improve the nutrient intake of institutionalized elderly [24]. On the contrary, Wright *et al.* reported that the elderly patients on a texture-modified food diet have a lower intake of energy and protein than those taking a normal hospital diet [33]. Although research backgrounds were different in these studies, appropriate swallowing assessments and nutritional management will be necessary to induce modified foods.

### 4.5 Causative organisms of AP and treatments

Based on the studies in the 1970s, anaerobic organisms were presumed to predominantly cause AP [34, 35]. Because the previous studies often performed trans-tracheal sampling and assessed patients late in their clinical course [4, 34, 35], the contribution of anaerobes in AP

had been overestimated. Nowadays, the concept of AP has changed, and the involvement of anaerobic bacteria in CAP and HAP is considered to be smaller than expected. The usual causative organisms of CAP, e.g., *S. pneumoniae* and *S. aureus*, are thought to play a major role in AP etiology [36–38]. In accordance with previous studies, causative organisms in our AP patients in CAP were *S. aureus* (19.7%) and *Klebsiella* spp. (16.4%), and typical anaerobes of AP, e.g., *Fusobacterium* spp. and *Bacteroides* spp., were rare in this study. In NHCAP/HAP, causative organisms were not significantly different between AP and non-AP patients. This result may reflect the possibility that AP may be included in the category of non-AP in NHCAP/HAP cases. Considering the difficulty of accurate diagnose of AP, it will be increasingly important to perform swallowing assessments for correct diagnosis.

Regarding the use of the antibiotics, CTRX (37.3%), SBT/ABPC (36.5%), and LVFX (11.1%), the standard choices for empiric therapy of CAP in adults [39], were frequently used in non-AP treatments of CAP: this implies targets for the treatment of non-AP patients in this study were *S. pneumoniae* and *Haemophilus influenzae*, typical causative organisms of CAP [40]. SBT/ABPC was also frequently used in the treatment of AP in CAP (64.5%), and this antibiotic is recommended for most AP patients of CAP and HAP with a low risk of drug-resistant bacteria [1, 41]. In NHCAP/HAP, the selection of the antibiotics was not significantly different between AP and non-AP patients. We suspect that the selection of antibiotics in NHCAP/HAP was performed considering the possibility of AP, even in the diagnosis of non-AP. Collectively, in this study, the tendency of empiric therapies in pneumonia patients seems generally appropriate at the present time. Novel understanding of microbiology and pathology of AP has evolved with the use of advanced technology, such as targeted polymerase-chain-reaction, sequencing of bacterial 16S ribosomal RNA genes, and metagenomics [1], and new candidate microbes of AP have been identified in the oral cavity [42]. Further studies will be needed to clearly understand the actual pathogens of AP and select appropriate antibiotics.

## 5. Limitation

Our study is a retrospective observational study, using clinical records of pneumonia patients in all departments of eight hospitals. Patients in this study were treated not only by pulmonologists but also by non-pulmonologists. Although this limitation of difference in therapist qualification might cause prognostic differences in pneumonia patients, Komiya *et al.* reported that the overall prognosis of pneumonia in elderly patients might not necessarily improve, irrespective of treatment by pulmonologists [43]. The results of this study cannot apply to non-hospitalized pneumonia patients because only hospitalized patients were enrolled. Mortality detection was limited to the inpatient setting, and we did not assess the presence of "do not resuscitate" order. Further studies, including chronic care hospitals and nursing homes, will be required to reveal the overview of AP in the future.

The strengths of this study are its large size and the multicenter, multiregional population, including both urban and rural areas. Overall, we believe that a large number of pneumonia patients and variability of clinicians involved in this study could increase the generalization of the research, compensate for the limitations of the study and provide knowledge about the clinical differences between AP and non-AP patients.

## 6. Conclusion

AP accounts for 38.4% of all pneumonia cases, and for 42.8% of elderly cases in acute care hospitals in Northern Japan. Lower BMI, lower C-reactive protein, a lower ratio of homestay before hospitalization, a higher complication rate of cerebrovascular disease, dementia, and neuromuscular disease were significant characteristics of AP patients compared with non-AP

patients. Swallowing intervention may be associated with lowering mortality in AP patients. However, swallowing assessments, especially video endoscopy and video fluorography, are still not popular even in AP patients. With the growth of the elderly population, preventive methods such as swallowing intervention, e.g., swallowing training and oral care [44–47], pharmacologic therapy [48], and sarcopenia [49], and nutrition management [50] will become more critical. All the medical doctors involved in geriatric treatment, not only pulmonologists, need to develop a better understanding of AP.

## Supporting information

**S1 Data.**
(XLSX)

## Acknowledgments

The authors thank Dr. Hajime Kurosawa, professor of the department of occupational medicine, Tohoku university graduate school of medicine, for critical reading of this paper.

## Author Contributions

**Conceptualization:** Jun Suzuki, Ai Hirano-Kawamoto, Jun Ohta.

**Data curation:** Kengo Kato, Jun Ohta, Rei Kawata, Tomonori Kanbayashi, Masaki Hatano, Tadahisa Shishido, Yuya Miyakura, Kento Ishigaki, Yasunari Yamauchi, Miho Nakazumi, Takuya Endo, Hiroki Tozuka, Shiori Kitaya, Yuki Numano, Shotaro Koizumi, Yutaro Saito, Mutsuki Unuma, Eiichi Ishida, Toshiaki Kikuchi.

**Formal analysis:** Jun Suzuki, Yuta Kobayashi, Akira Ohkoshi.

**Funding acquisition:** Yukio Katori.

**Investigation:** Ryoukichi Ikeda, Rei Kawata, Tomonori Kanbayashi, Masaki Hatano, Tadahisa Shishido, Yuya Miyakura, Kento Ishigaki, Yasunari Yamauchi, Miho Nakazumi, Takuya Endo, Hiroki Tozuka, Shiori Kitaya, Yuki Numano, Shotaro Koizumi, Yutaro Saito, Mutsuki Unuma, Eiichi Ishida, Toshiaki Kikuchi.

**Project administration:** Ryoukichi Ikeda, Kengo Kato, Risako Kakuta, Jun Ohta, Yukio Katori.

**Supervision:** Ryoukichi Ikeda, Ken Hashimoto, Takayuki Kudo, Kenichi Watanabe, Masaki Ogura, Masaru Tateda, Takatsuna Sasaki, Nobuo Ohta, Tatsuma Okazaki, Yukio Katori.

**Validation:** Nobuo Ohta, Tatsuma Okazaki, Yukio Katori.

**Writing – original draft:** Jun Suzuki, Ryoukichi Ikeda.

**Writing – review & editing:** Ryoukichi Ikeda, Risako Kakuta, Akira Ohkoshi, Ryo Ishii, Ai Hirano-Kawamoto, Jun Ohta, Ken Hashimoto, Takayuki Kudo, Kenichi Watanabe, Masaki Ogura, Masaru Tateda, Takatsuna Sasaki, Nobuo Ohta, Tatsuma Okazaki.

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
