## [Decision Letter · Decision Letter 0]

28 Apr 2021

PONE-D-21-00691

Characteristics of aspiration pneumonia patients in acute care hospitals: a multicenter, retrospective survey in Northern Japan

PLOS ONE

Dear Dr. Ikeda,

Thank you for submitting your manuscript to PLOS ONE. After careful consideration, we feel that it has merit but does not fully meet PLOS ONE’s publication criteria as it currently stands. Therefore, we invite you to submit a revised version of the manuscript that addresses the points raised during the review process.

Your manuscript was assessed by two external experts who have raised several important concerns about the methodology and statistical analyses used, as well as the extent to which the conclusions as written are supported by the results.

.

We look forward to receiving your revised manuscript.

Kind regards,

Dr Joseph Donlan

Senior Editor

PLOS ONE

Journal Requirements:

"This research was supported by AMED under Grant Number 19dk0310101h0001."

"No"

"No"

7. Your ethics statement should only appear in the Methods section of your manuscript. If your ethics statement is written in any section besides the Methods, please delete it from any other section.

8.  We note that Figure 1 in your submission contain map images which may be copyrighted. All PLOS content is published under the Creative Commons Attribution License (CC BY 4.0), which means that the manuscript, images, and Supporting Information files will be freely available online, and any third party is permitted to access, download, copy, distribute, and use these materials in any way, even commercially, with proper attribution. For these reasons, we cannot publish previously copyrighted maps or satellite images created using proprietary data, such as Google software (Google Maps, Street View, and Earth). For more information, see our copyright guidelines: http://journals.plos.org/plosone/s/licenses-and-copyright.

8.1.    You may seek permission from the original copyright holder of Figure 1 to publish the content specifically under the CC BY 4.0 license. 

8.2.    If you are unable to obtain permission from the original copyright holder to publish these figures under the CC BY 4.0 license or if the copyright holder’s requirements are incompatible with the CC BY 4.0 license, please either i) remove the figure or ii) supply a replacement figure that complies with the CC BY 4.0 license. Please check copyright information on all replacement figures and update the figure caption with source information. If applicable, please specify in the figure caption text when a figure is similar but not identical to the original image and is therefore for illustrative purposes only.

9. Please include a separate caption for each figure in your manuscript.

Reviewers' comments:

Reviewer's Responses to Questions

**Comments to the Author**

1. Is the manuscript technically sound, and do the data support the conclusions?

Reviewer #1: Partly

Reviewer #2: Partly

2. Has the statistical analysis been performed appropriately and rigorously? 

Reviewer #1: No

Reviewer #2: Yes

3. Have the authors made all data underlying the findings in their manuscript fully available?

Reviewer #1: Yes

Reviewer #2: Yes

4. Is the manuscript presented in an intelligible fashion and written in standard English?

Reviewer #1: Yes

Reviewer #2: Yes

5. Review Comments to the Author

Reviewer #1: The paper is primarily descriptive. However, the authors do use statistics such as The Student's t-test, the Wilcoxon–Mann–Whitney test, or Fisher's exact test. All analyses are univariate. Tables 1 to 5 list many clinical and pathologic characteristics of the study sample. No attempt has been made at any statistical sophistication examining the variables of interest in a multivariate framework considering the confounding factors on any well defined endpoints.

For example the authors note that the present study revealed that swallowing interventions significantly changed the dietary pattern from a normal diet to a modified diet. Also they state that malnutrition is common in elderly people, especially those with chronic disorders , and is associated with poor hospitalization outcomes. Some adjustment of odds ratios through routine logistic analyses could have been attempted. The manuscript should be re-examined from the statistical perspective with appropriate adjustment of the important variables of interest on perhaps differentiating between AP and non AP pneumonia. The sample size is certainly large enough.

Reviewer #2: In, “Characteristics of aspiration pneumonia patients in acute care hospitals: a multicenter, retrospective survey in Northern Japan,” Suzuki and colleagues characterize hospitalized adults with pneumonia, comparing aspiration pneumonia to non-aspiration pneumonia.

Major:

1. Abstract, Results: “Swallowing intervention improved in-hospital mortality.” This study is not designed to allow such an inference. Please revise to, “Swallowing intervention was associated with lower in-hospital mortality.”

2. The microbiologic comparison of aspiration and non-aspiration pneumonia is methodologically flawed. These cohorts have different composition with respect to nursing-home or hospital status. The correct comparison should really be community acquired aspiration pneumonia vs. community-acquired non-aspiration pneumonia, and nursing home/healthcare-associated aspiration pneumonia vs. nursing home/healthcare-associated non-aspiration pneumonia. It is unsurprising that table 4 demonstrates different microbiological patterns. What would be more interested is whether aspiration patients have any different microbiologic patterns that transcend their designation to community, hospital, or nursing home. Similarly, the antibiotic patterns are not as informative when aspiration pneumonia has a higher proportion of NHCAP. After doing these analyses, the discussion should also be revised accordingly.

3. Section 4.4 is a substantial leap from the data. There are several other possible explanations for this association, including confounding by indication, selection bias, etc. While it is an interesting speculation, and may be true that swallowing intervention might reduce mortality, your study cannot demonstrate this. Please revise this section to avoid overstating conclusions and restrict the discussion to avoid unsubstantiated claims

Minor:

1. Line 221: “in patients” is written with strikethrough.

6. PLOS authors have the option to publish the peer review history of their article (what does this mean?). If published, this will include your full peer review and any attached files.

Reviewer #1: No

Reviewer #2: No

---

## [Author Response · Author response to Decision Letter 0]

10 Jun 2021

Journal Requirements:

Q1. Please ensure that your manuscript meets PLOS ONE's style requirements, including those for file naming. The PLOS ONE style templates can be found at

A1. OK

Q2. Please provide additional details regarding participant consent. In the ethics statement in the Methods and online submission information, please ensure that you have specified (1) whether consent was informed and (2) what type you obtained (for instance, written or verbal, and if verbal, how it was documented and witnessed). If your study included minors, state whether you obtained consent from parents or guardians. If the need for consent was waived by the ethics committee, please include this information.

A2. All data were fully anonymized, and IRB in our institution waived the requirement for informed consent.

Q3. Thank you for stating the following in the Funding Section of your manuscript:

"This research was supported by AMED under Grant Number 19dk0310101h0001."

"No"

A3. We have removed the funding-related text from the manuscript and request you to update our Funding Statement as follows: This research was supported by AMED under Grant Number 19dk0310101h0001. We have also included our amended statements within our cover letter.

Q4. Thank you for stating the following in your Competing Interests section: 

"No"

A4. We request you to update our Competing Interests on the online submission form as follows: The authors have declared that no competing interests exist. We have also included our amended statements within our cover letter.

Q5. In your Data Availability statement, you have not specified where the minimal data set underlying the results described in your manuscript can be found. PLOS defines a study's minimal data set as the underlying data used to reach the conclusions drawn in the manuscript and any additional data required to replicate the reported study findings in their entirety. All PLOS journals require that the minimal data set be made fully available. For more information about our data policy, please see http://journals.plos.org/plosone/s/data-availability.

A5. We have uploaded our minimal underlying data set as Supporting Information files.

Q6. We note that you have included the phrase “data not shown” in your manuscript. Unfortunately, this does not meet our data sharing requirements. PLOS does not permit references to inaccessible data. We require that authors provide all relevant data within the paper, Supporting Information files, or in an acceptable, public repository. Please add a citation to support this phrase or upload the data that corresponds with these findings to a stable repository (such as Figshare or Dryad) and provide and URLs, DOIs, or accession numbers that may be used to access these data. Or, if the data are not a core part of the research being presented in your study, we ask that you remove the phrase that refers to these data.

A6. We have removed the phrase “data not shown” in our manuscript and revised the corresponding sentence.

Q7. Your ethics statement should only appear in the Methods section of your manuscript. If your ethics statement is written in any section besides the Methods, please delete it from any other section.

A7. Accordingly, we have removed the other ethical statement apart from the Methods section.

Q8. We note that Figure 1 in your submission contain map images which may be copyrighted. All PLOS content is published under the Creative Commons Attribution License (CC BY 4.0), which means that the manuscript, images, and Supporting Information files will be freely available online, and any third party is permitted to access, download, copy, distribute, and use these materials in any way, even commercially, with proper attribution. For these reasons, we cannot publish previously copyrighted maps or satellite images created using proprietary data, such as Google software (Google Maps, Street View, and Earth). For more information, see our copyright guidelines: http://journals.plos.org/plosone/s/licenses-and-copyright.

A8. We have included maps that we created ourselves, so no copyright permission is required.

Q9. Please include a separate caption for each figure in your manuscript.

A9. We have added separate captions in Fig 1.

 

Reviewers' comments:

Reviewer #1: 

The paper is primarily descriptive. However, the authors do use statistics such as The Student's t-test, the Wilcoxon–Mann–Whitney test, or Fisher's exact test. All analyses are univariate. Tables 1 to 5 list many clinical and pathologic characteristics of the study sample. No attempt has been made at any statistical sophistication examining the variables of interest in a multivariate framework considering the confounding factors on any well defined endpoints.

For example the authors note that the present study revealed that swallowing interventions significantly changed the dietary pattern from a normal diet to a modified diet. Also they state that malnutrition is common in elderly people, especially those with chronic disorders, and is associated with poor hospitalization outcomes. Some adjustment of odds ratios through routine logistic analyses could have been attempted. The manuscript should be re-examined from the statistical perspective with appropriate adjustment of the important variables of interest on perhaps differentiating between AP and non AP pneumonia. The sample size is certainly large enough.

Answers:

Thank you for your suggestion, and we appreciate the time and effort to provide insightful feedback to our paper. We agree with you. We have performed logistic regression analyses as to the characteristics of AP patients and swallowing intervention groups of AP patients. We showed that lower BMI, lower C-reactive protein, a lower ratio of homestay before hospitalization, a higher complication rate of cerebrovascular disease, dementia, and neuromuscular disease are significant characteristics of AP patients and that a lower ratio of AP recurrence, lower ratio of homestay before hospitalization, higher rate of swallowing assessments are significant characteristics of swallowing intervention cases in AP patients. We are grateful that the results have become more evident thanks to your comments.

Reviewer #2: In, “Characteristics of aspiration pneumonia patients in acute care hospitals: a multicenter, retrospective survey in Northern Japan,” Suzuki and colleagues characterize hospitalized adults with pneumonia, comparing aspiration pneumonia to non-aspiration pneumonia.

Major:

Q1. Abstract, Results: “Swallowing intervention improved in-hospital mortality.” This study is not designed to allow such an inference. Please revise to, “Swallowing intervention was associated with lower in-hospital mortality.”

A1. Thank you for your suggestion and, we appreciate the time and effort to provide insightful feedback to our paper. We have reflected this comment by removing the sentence “Swallowing intervention improved in-hospital mortality.” We have revised the sentences related to this point in the Abstract, Discussion, and Conclusion. 

Q2. The microbiologic comparison of aspiration and non-aspiration pneumonia is methodologically flawed. These cohorts have different composition with respect to nursing-home or hospital status. The correct comparison should really be community acquired aspiration pneumonia vs. community-acquired non-aspiration pneumonia, and nursing home/healthcare-associated aspiration pneumonia vs. nursing home/healthcare-associated non-aspiration pneumonia. It is unsurprising that table 4 demonstrates different microbiological patterns. What would be more interested is whether aspiration patients have any different microbiologic patterns that transcend their designation to community, hospital, or nursing home. Similarly, the antibiotic patterns are not as informative when aspiration pneumonia has a higher proportion of NHCAP. After doing these analyses, the discussion should also be revised accordingly.

A2. We agree with you. We have performed a microbiologic comparison as you suggested: AP in CAP vs. non-AP in CAP, and AP in NHCAP/HAP vs. non-AP in NHCAP/HAP. We believe that the characteristics of causative organisms and antibiotic selection become more precise thanks to your suggestion. We have added some sentences in the Discussion section according to the new results. 

Q3. Section 4.4 is a substantial leap from the data. There are several other possible explanations for this association, including confounding by indication, selection bias, etc. While it is an interesting speculation, and may be true that swallowing intervention might reduce mortality, your study cannot demonstrate this. Please revise this section to avoid overstating conclusions and restrict the discussion to avoid unsubstantiated claims

A3. We thank you for raising an important suggestion. The result of a significant decrease in in-hospital mortality in the swallowing intervention group was not supported in multivariate analysis. However, considering the limitation of this retrospective study, we speculate that swallowing intervention might reduce mortality. We have revised this section not to overstate conclusions. 

Minor:

Q1. Line 221: “in patients” is written with strikethrough.

A1. We have revised the sentence.

Other points revised:

We have added Yuta Kobayashi, Akira Ohkoshi, and Ryo Ishii, who contributed to the multivariate analyses to the authors.

We have revised Figure 1 to deal with copyright issues.

We have changed “PaO2” to “SpO2” in Tables 1 and 3.

We have added Tables 2 and 4, results of multivariate analyses.

---

## [Decision Letter · Decision Letter 1]

24 Jun 2021

Characteristics of aspiration pneumonia patients in acute care hospitals: a multicenter, retrospective survey in Northern Japan

PONE-D-21-00691R1

Dear Dr. Ikeda,

We’re pleased to inform you that your manuscript has been judged scientifically suitable for publication and will be formally accepted for publication once it meets all outstanding technical requirements.

Kind regards,

Michael J Lanspa, MD

Guest Editor

PLOS ONE

Additional Editor Comments (optional):

Reviewers' comments:

Reviewer's Responses to Questions

**Comments to the Author**

1. If the authors have adequately addressed your comments raised in a previous round of review and you feel that this manuscript is now acceptable for publication, you may indicate that here to bypass the “Comments to the Author” section, enter your conflict of interest statement in the “Confidential to Editor” section, and submit your "Accept" recommendation.

Reviewer #1: All comments have been addressed

2. Is the manuscript technically sound, and do the data support the conclusions?

Reviewer #1: (No Response)

3. Has the statistical analysis been performed appropriately and rigorously? 

Reviewer #1: (No Response)

4. Have the authors made all data underlying the findings in their manuscript fully available?

Reviewer #1: (No Response)

5. Is the manuscript presented in an intelligible fashion and written in standard English?

Reviewer #1: (No Response)

6. Review Comments to the Author

Reviewer #1: (No Response)

7. PLOS authors have the option to publish the peer review history of their article (what does this mean?). If published, this will include your full peer review and any attached files.

Reviewer #1: No

---

## [Editor Report · Acceptance letter]

21 Jul 2021

PONE-D-21-00691R1 

Characteristics of aspiration pneumonia patients in acute care hospitals: a multicenter, retrospective survey in Northern Japan 

Dear Dr. Ikeda:

I'm pleased to inform you that your manuscript has been deemed suitable for publication in PLOS ONE. Congratulations! Your manuscript is now with our production department. 

Kind regards, 

on behalf of

Dr. Michael J Lanspa 

Guest Editor

PLOS ONE